# Prevalence of Untreated Early Childhood Caries of 5-Year-Old Children in Hong Kong: A Cross-Sectional Study

**DOI:** 10.3390/ijerph182211934

**Published:** 2021-11-13

**Authors:** Faith Miaomiao Zheng, Iliana Gehui Yan, Duangporn Duangthip, Sherry Shiqian Gao, Edward Chin Man Lo, Chun Hung Chu

**Affiliations:** 1Faculty of Dentistry, The University of Hong Kong, Hong Kong 999077, China; zhengmm@connect.hku.hk (F.M.Z.); iliana@connect.hku.hk (I.G.Y.); dduang@hku.hk (D.D.); hrdplcm@hku.hk (E.C.M.L.); 2Department of Stomatology, School of Medicine, Xiamen University, Xiamen 361000, China; sherrysgao@xmu.edu.cn

**Keywords:** children, caries, prevention, kindergarten, outreach

## Abstract

This cross-sectional survey investigated untreated early childhood caries (ECC) and its associated factors among 5-year-old children in Hong Kong. Children were recruited using a multistage sampling method. One dentist examined the children in kindergarten to diagnose untreated ECC (dt) at the cavitation level. Each child’s demographic information, snacking behaviour, and oral health-related practice were collected using a parental questionnaire. The relationships between the untreated ECC and demographic information, snacking behaviours, and oral health-–related practice were analysed by zero-inflated negative binomial (ZINB) regression analysis. This survey recruited 404 children. Their dt score was 2.8 ± 3.8. The significant untreated ECC (SiUC) index, which was one-third of the children with the highest dt score, was 7.1 ± 3.6. Their untreated ECC prevalence was 57%, which was associated with the district the child lived in. Most children with untreated ECC (71%, 164/231) had never visited a dentist. Children who brushed without toothpaste had more untreated ECC. Children coming from low-income families and with a lower maternal education level had a higher risk of ECC. In conclusion, untreated ECC was prevalent and unevenly distributed among 5-year-old children in Hong Kong. Its prevalence was associated with toothpaste use, family income, maternal education level and the district they lived in.

## 1. Introduction

Dental caries or early childhood caries (ECC) is a common oral disease in preschool children worldwide [1]. The American Academy of Paediatric Dentistry defined ECC as the presence of one or more decayed, missing or filled tooth surfaces in any primary tooth in children 71 months of age or younger [2]. The aetiological factors of EEC include frequent sugar consumption in an environment of enamel adherent bacteria and the mechanism of sustained acid production and resulting demineralization of teeth structures [3]. The prevalence of ECC is the result of a variety of factors such as culture, ethnicity, socioeconomic status, lifestyle, dietary pattern, and oral hygiene practice. A report of the International Association of Paediatric Dentistry in 2018 showed that the overall ECC prevalence of 5-year-old children from 72 studies worldwide was 63% [4]. More than 530 million children worldwide had ECC in their primary teeth, and most ECC cases remain untreated [5].

The impact of untreated ECC is not just pain and infection, but about more immediate and long-term consequences such as an increased risk of new carious lesions, extraction of primary teeth and malocclusion with various clinical implications such as deep overbite, midline deviation, excessive overjet, anterior cross-bite and malalignment. It not only affects oral health but also general health. ECC can adversely affect children’s growth and development and their well-being [3].

The Department of Health of Hong Kong conducted oral health surveys on 6-year-old children when water fluoridation started in 1961. In 1997, the World Health Organization (WHO) recommended that researchers carry out a survey for children between their fifth and sixth birthday (i.e., 5-year-olds) [6]. This age is of interest in relation to caries status in the primary dentition, which can exhibit changes over a short time span. Since then, the Department of Health conducted two oral health surveys on 5-year-old children in 2001 and 2011. Both surveys found that the prevalence of ECC in 5-year-old children was 51% [7]. Apart from the Department of Health, The University of Hong Kong also conducted and published several oral health surveys for 5-year-old children in Hong Kong (Appendix A).

They published their first survey in 1986. The survey found that the prevalence of ECC among 5-year-old children was 63% [8]. They performed subsequent surveys in 1997, 2007, 2009 and 2016. These surveys reported that approximately half of the 5-year-old children suffered from ECC and that more than 90% of their ECCs were left untreated [9,10,11,12]. These surveys suggested that the ECC status of Hong Kong preschool students remained high and that their oral health did not benefit from rapid economic growth in the same period. However, there was no organized preventive/therapeutic programme for preschool children.

Because of the high prevalence of ECC among preschool children, a charity organization funded the University of Hong Kong to provide an outreach dental service to all kindergarten children in 2019. This service provided dental screening to identify children with ECC and provided silver diamine fluoride therapy to arrest ECC with parental consent [13]. The service also provided oral health education to children’s parents and training to kindergarten teachers in order to deliver oral health messages to students for the prevention of ECC. This survey dovetailed to the outreach service. Previous studies have focused on caries experience and the relationship between the common caries risk factors and ECC experience [9,11,12]. These studies also found that most of the ECC cases were left untreated. The objective of this study is to investigate the prevalence of untreated ECC and to identify factors associated with untreated ECC among 5-year-old kindergarten children in Hong Kong.

## 2. Materials and Methods

The report of this survey was followed the Statement of Strengthening the Reporting of Observational Studies in Epidemiology (STROBE) [14].

### 2.1. Study Participnts

This study was conducted in 2019 to research untreated ECC among 5-year-old children. The 5-year-old children were invited to join this survey at the kindergarten level. The children and their parents were not incentivized to participate in this study. The study consisted of a questionnaire survey for the participating children’s parents and a clinical examination of the participating children in kindergarten. Children with serious health problems, such as congenital health disease, were excluded from study.

### 2.2. Sampling Method

This survey used a multistage sampling method to recruit the 5-year-old children. Hong Kong is divided into three districts, namely Hong Kong Island, Kowloon and the New Territories. The number of children invited from each of the three districts was determined according to the population-to-district ratio [15]. Registered kindergartens were chosen by the random sampling method, and all 5-year-old children in the selected kindergartens were invited to participate in the survey.

### 2.3. Sample Size Calculation

Based on the survey performed in 2016, this survey estimated the prevalence of untreated ECC (dt) to be 52% [12]. The confidence level was set at 95%, and the confidence interval (CI) was set at 5% (CI: 47–57%). The sample size required was 383 children [14]. This survey aimed to recruit 400 children, with 50 children from Hong Kong Island, 120 from Kowloon and 230 from the New Territories, according to the population-to-district ratio.

### 2.4. Questionnaire Survey

A validated questionnaire used in previous oral health surveys was sent to the participating children’s parents [11,12] through the selected kindergarten. It consisted of three parts, namely, (i) The child’s and family’s information: sex, date of birth and birthplace; (ii) The child’s oral health-related behaviours: toothbrushing and sugary snacking habits; and (iii) The child’s family income and the parents’ education levels. An assistant checked all returned questionnaires and followed up on any missing, unclear or inappropriate answers on the questionnaire by phone.

### 2.5. Clinical Examination

One trained dentist performed the clinical examination in a kindergarten classroom using a disposable dental mirror with a light-emitting diode and ball-end Community Periodontal Index probe. The diagnostic criteria for caries followed the WHO’s recommendations [6]. Food debris was gently removed for the diagnosis of ECC at the cavitation level, which is the same as the decayed (dt) index. The examiner was calibrated and her intra-examination agreement of caries diagnosis had to achieve a Kappa value of at least 90% before conducting this survey. The recalibration was conducted every 6 months. Intra-examiner agreement was monitored by re-examining a 10% random sample of the children in each visit. A dental assistant randomly selected the children for the examiner to perform a duplicated examination without notifying the examiner. After the dental examination, the dentist provided the parents with the child’s individual oral health report.

### 2.6. Statistical Analysis

Data analysis was performed using Statistical Package for Social Science version 27.0 (SPSS Inc., Chicago, IL, USA), Stata version 14 (StataCorp, College Station, TX, USA). The intra-examiner agreement was evaluated using Kappa statistics. The SiUC index was calculated by the mean of the untreated ECC (dt) of the one-third of the children with the highest dt score [16]. SiUC is used as a complement to the mean dt value. A chi-square test was used to test the association of an untreated ECC index with the variables studied. The Mann–Whitney U or Kruskal–Wallis H test with Bonferroni adjusted pairwise comparison was employed to study the distribution of dt scores according to different variables.

Independent variables included mother’s education level, father’s education level, monthly family income, using toothpaste for tooth brushing, toothbrushing frequency, snacking frequency and primary caregivers. Independent variables with *p*-values less than 0.10 for a chi-square test, Mann–Whitney U test and Kruskal–Wallis H test were chosen as selected variables in the regression model. The Poisson model, negative binomial model and zero-inflated model with robust standard error were performed to study the relationship between the dt scores and the selected variables. Vuong’s test was employed to determine the final model for analysis. Backward stepwise selection was used to remove the selected variables with a *p*-value equal or larger than 0.05 from the regression model. The level of significance for all tests was set at 0.05.

### 2.7. Research Ethics and Study Participnts

Ethical approval for the study was sought from the Institutional Review Board of HKU/Hospital Authority Hong Kong West Cluster (IRB UW 16-180). Written consent was obtained from the parents before their children joined the survey.

## 3. Results

### 3.1. Overall Situation of Untreated ECC among Hong Kong 5-Year-Old Children

A total of 404 children completed this survey. Among the children in this study, 216 (65%) were boys. The mean dt score (±SD) was 2.8 ± 3.8. The distribution of dt score was positively skewed, with a skewness of 1.6 (Figure 1). Untreated ECC (dt > 0) was found in 231 (57%) children. Among those children with untreated ECC, 117 (29%) had 1 to 3 untreated caries teeth. The SiUC was 7.1. One child had the highest untreated ECC with 19 caries teeth. The Kappa value for intra-examiner agreement on untreated ECC diagnosis was 0.96.

### 3.2. Distribution of Untreated ECC Teeth

Figure 2 shows the distribution of untreated ECC in the children. Upper central incisors had the highest prevalence of untreated ECC (33%), followed by lower first and lower second molars. Lower canine had the lowest prevalence of untreated ECC (2%). It is noteworthy that the lower central incisor had an untreated ECC prevalence of 10%. Anterior teeth (54%) and upper teeth (57%) had more untreated ECC than their counterparts.

### 3.3. Untreated ECC Prevalence According to District, Birthplace and Sex

Table 1 summarises the untreated ECC prevalence according to district, birthplace and sex. The results indicated that these preschool students among the three districts they lived in are associated with the untreated ECC prevalence (*p* < 0.003), dt distribution (*p* = 0.002) and SiUC index (*p* < 0.001). The dt distribution and mean SiUC score of the children who lived in Hong Kong Island was lower than those in Kowloon and New Territories. Although our results are most compatible with no important association between untreated ECC prevalence and children born in mainland China or not born in mainland China (*p* = 0.174), children born in mainland China had a higher mean SiUC score than those who were not born in mainland China (*p* < 0.001). In addition, our results are most compatible with no important association between children’s sex and untreated ECC prevalence (*p* = 0.480), mean rank of dt (*p* = 0.256) or mean SiUC index (*p* = 0.369).

### 3.4. Untreated ECC and Its Associated Factors among 5-Year-Old Hong Kong Children

Table 2 summarises the association of untreated ECC prevalence (dt > 0) and independent variables. Children whose fathers or mothers had received secondary education or below had higher untreated ECC prevalence than those whose parents had a post-secondary education level (*p* = 0.004, *p* < 0.001). Furthermore, a higher proportion of children with untreated ECC was found in families with a monthly income of less than HK 15,000 (*p* < 0.001). Children who brushed their teeth with toothpaste had lower untreated ECC prevalence compared to those who brushed their teeth without toothpaste (*p* = 0.032).

This survey’s results are most compatible with no important association between untreated ECC prevalence among children and whether they brushed their teeth twice a day (*p* = 0.120), whether they snacked more than twice a day (*p* = 0.090) or whether their parents were primary caregivers (*p* = 0.740). We also found that most of the children (361/404, 89%) brushed their teeth without toothpaste and that most of their primary caregivers were their parents (287/404, 71%). About two-thirds of the children (263/404, 65%) brushed their teeth at least twice daily. Fifty-three children (13%) had sugary snacks more than twice daily. The majority of children (307/404, 76%) had never visited a dentist. Among the 231 children who had untreated ECC, 164 of them (164/231, 71%) had never visited a dentist (which is not shown in Table 2).

The Mann–Whitney U and Kruskal–Wallis H tests showed a higher dt distribution for children who did not use toothpaste when brushing (*p* = 0.029), ate sugary snacks more than twice daily (*p* = 0.040), had lower parental education levels (mother: *p* < 0.001, father: *p* = 0.040) or had lower family monthly income (*p* < 0.001). Our results are most compatible (*p* = 0.920) with no important association between dt scores and children whose parents were their primary caregivers or those whose primary caregivers were not their parents. Table 3 shows untreated ECC and independent variables.

Vuong’s test and likelihood test showed that the ZINB model fit the data better than the standard negative binomial (*p* < 0.001), and the likelihood test showed that the ZINB model fit the data better than the zero-inflated Poisson (ZIP) model (*p* < 0.001). Table 4 shows the final model of the ZINB analysis of untreated ECC and risk factors.

The results from the negative binomial part indicated that family income was negatively associated with the dt score for a monthly income ranging from HK 15,001 to HK 30,000 (*p* = 0.093, incidence risk ratio [IRR]: 0.801, 95% CI: 0.619, 1.037) and for a monthly income over HK 30,000 (*p* = 0.005, IRR: 0.589, 95% CI: 0.408, 0.852) compared to a monthly income below HK 15,000, respectively. In the zero-inflated model, among children from a family with a monthly income over HK 30,000, we found that the chance of having ‘no untreated ECC’ (i.e., being ‘an excess zero’) was 3.455 times as likely (*p* = 0.003, 95% CI: 1.524, 7.830) compared to those from a family with a monthly income below HK 15,000. The results from the zero-inflated part indicated that children brushing their teeth with toothpaste were 2.362 times as likely (*p* = 0.044, 95% CI: 1.022, 5.463) to have ‘no untreated ECC’ compared to those who brush their teeth without toothpaste. This model also shows that children whose mothers graduated from post-secondary school had a higher chance of having ‘no untreated ECC’ (*p* = 0.007, odds ratio: 2.211, 95% CI: 1.238, 3.949) than those whose mothers had received secondary education or below.

## 4. Discussion

This study collected the data of untreated ECC among 5-year-old preschool children to monitor the dental outreach service, which is offered free of charge to all kindergartens in Hong Kong. We followed the Statement of Strengthening the Reporting of Observational Studies in Epidemiology (STROBE) in our reporting of this survey [14]. We selected the children by random sampling in baseline, and they did not receive the dental care outreach service. To evaluate the dental outreach service, we compared their ECC status to those children of the same age who had received the dental care outreach service after 36 months.

Sociodemographic background is one of the common factors affecting the oral health of preschool children [12]. This study used the multistage sampling method based on various districts to collect the data of untreated ECC because our previous studies showed that the three districts had different sociodemographic backgrounds [9,11,12]. Multistage sampling divided the child population into districts, and this sampling method made the between-district variance low and the within-district variance high.

Children living in one district could have a relatively homogenous background, whereas the children living in other districts might have somewhat different backgrounds. Therefore, this cluster sample by district could give a larger variance than a simple random sample, and the estimates could be less precise. The multistage sampling method is generally simple and flexible. However, this sampling method could not provide 100% representation of the population. It omitted portions of the population from the study, and data could have been lost in the sense that not everyone was counted.

This survey recruited 5-year-old children based on kindergartens because it saves the time, cost and efforts associated with the survey. Recruiting preschool children based on kindergartens is largely appropriate because the government subsidizes preschool education through kindergarten [17]. Most 5-year-old children in Hong Kong attend kindergartens, whereas home schooling is uncommon. We randomly selected kindergartens based on different districts, namely, four kindergartens from New Territories, three kindergartens from Kowloon and one school from Hong Kong Island. Finally, the ratio of recruited children in Hong Kong Island, Kowloon and New Territories was 5:12:23, which was the population ratio of the three districts. Therefore, statistical sample weighting was not necessary in this study.

The diagnostic criteria in this study are the same as the dt of the dmft index, which is a commonly used index for epidemiological studies. The clear and distinct diagnostic criteria at the cavitation level allowed a very high Kappa value in the intra-examiner agreement. However, carious lesions, particularly those in the proximal surface, were not recorded according to these diagnostic criteria. Similar to other studies, the distribution of the dt index in this study did not follow normal distribution. Therefore, we used the Mann–Whitney U and Kruskal–Wallis H tests, which are non-parametric methods for data analysis.

Similar to other studies, the distribution of dt scores in this study was over-dispersed with ‘excess zero’ [18]. We used ZINB regression analysis in this study because ZINB regression analysis can be used to analyse a fixed set of covariate values such as the dt index. The Vuong’s and likelihood-ratio tests showed that the ZINB regression model saw better fit with this study’s data compared to the ZIP regression model and standard negative binomial regression model. ZINB regression analysis includes two portions, namely the zero-inflated portion and negative binomial portion. The zero-inflated portion constituted a mixture of a standard probability distribution for counting data. The negative binomial portion typically represented a ‘susceptible’ subpopulation of children at risk for ECC. A subpopulation of ‘non-susceptible’ children with no untreated ECC was considered to not be at risk [18]. In the regression model, we analysed variables with a *p*-value under 0.10 instead of 0.05 as covariates. This protocol of analysis reduced the influence of irrelevant variables and prevented the omission of important factors.

Our previous survey found that upper central incisors, followed by lower second molars and lower first molars, were the most common teeth affected by ECC [11]. In this survey, upper central incisors had the highest prevalence of untreated ECC. The lower first molars had a higher prevalence of untreated ECC than lower second molars, which is consistent with previous discoveries [11]. The reason could be that the first molars erupted earlier than the second molars. Lower incisors in general are less likely to develop ECC due to the proximity of the submandibular salivary gland’s opening and the tongue’s physical protection. A study reported that only 2% to 5% of lower incisors developed ECC [19]. However, this survey found approximately 10% of the lower central incisors had untreated ECC. Although lower incisors have a lower ECC risk than other teeth, children with multiple ECC could have had untreated ECC on their lower incisors. This survey found that most children (70%) who had untreated carious lower incisors had untreated ECC on other teeth.

Literature found a discrepancy in the prevalence of ECC cases based on sex, and girls typically have a higher ECC prevalence than boys [20]. The higher ECC prevalence among girls is often attributed to the longer exposure of girls’ teeth to a cariogenic oral environment due to their earlier eruption of teeth. However, this study found that sex was not associated with untreated ECC. This finding is consistent with our previous surveys in the past two decades [9,11,12]. Unlike in Hong Kong, there is no water fluoridation in mainland China. ECC prevalence and experience among children in mainland China are higher than in Hong Kong [9,11]. Hong Kong is part of China, and over the past 20 years, an average of 131 new immigrants travel to Hong Kong every day, i.e., about 48,000 a year, mainly for family reunions [21]. A large proportion of these new immigrants is children. Our previous studies found that children born in the mainland had higher cases of ECC in 1999 and 2009 [9,11]. However, this survey found that the difference of untreated ECC between the children who were born in mainland China and those who were not born in mainland China was not as large as that in previous studies [9,11].

This study found that children who brushed with toothpaste had less untreated ECC. The WHO recommends that children aged 2–5 years should brush with toothpaste [22]. However, only 11% (43/404) of children use toothpaste to brush their teeth. Therefore, it is essential to promote the use of toothpaste in toothbrushing and encourage parents to brush for their child. Similar to our survey conducted in 2016, this study found that children who (i) consumed sugary snacks more than twice daily, (ii) had low monthly family income or (iii) had lower parental education levels had more untreated ECC [11]. Oral health promotion should be tailor-made and offer dietary analysis and advice to children’s parents. Parents should also control their child’s sugar intake. Snacks and beverages need to be labelled to indicate their sugar content in Hong Kong. Caries-prevention programmes should be targeted to children with a lower family income and to parents with a low education level.

The results of this study showed that the untreated prevalence of ECC among 5-year-old children in Hong Kong not only remains high but is higher than those surveys reported in the past 2 decades, compared to 40% in 1997 [9], 43% in 2007 [10], 47% in 2009 [11] and 51% in 2016 [12]. The COVID-19 pandemic profoundly affected our lives. Many dental treatments involve close contact with patients and aerosol generation. Aerosolized particles of viruses can be airborne and remain suspended in the air for hours, facilitating their transmission. Almost all dental service had to be suspended, and this adversely affected people’s oral health.

The WHO and the World Dental Federation in 2000 set the goal that at least half of preschool children should be caries-free [23]. However, Hong Kong has not achieved the goal because surveys have shown that more than half of the children have untreated ECC. The prevalence of ECC in Singapore is lower than that in Hong Kong. In addition, by 2009, Singapore achieved the goal of ensuring that more than half of children have no caries experience [24]. Singapore and Hong Kong have a similar economic development and cultural background. Singapore was the first Asian country to implement water fluoridation in 1958. Provision of free school-based dental care with an emphasis on prevention has played an important role in promoting good dental health among children in Singapore since 1970 [24]. Therefore, comprehensive dental care focusing on prevention is essential for preschool children in Hong Kong. This is particularly important because there has been a gradual increase in untreated ECC prevalence in the past two decades since 1997.

The high and increasing prevalence of untreated ECC warrants effective and organized dental care in order to promote and improve preschool children’s oral health. In 2019, The Hong Kong Jockey Club Charities Trust supported the University of Hong Kong in providing free outreach dental care to all 180,000 kindergarten children so as to prevent ECC among children in Hong Kong.

## 5. Conclusions

Most of Hong Kong’s 5-year-old children have never visited a dentist, and more than half of them have untreated ECC. The untreated ECC was unevenly distributed, and its prevalence was associated with the child’s use of toothpaste for brushing, family income and maternal education level.

## Figures and Tables

**Figure 1 ijerph-18-11934-f001:**
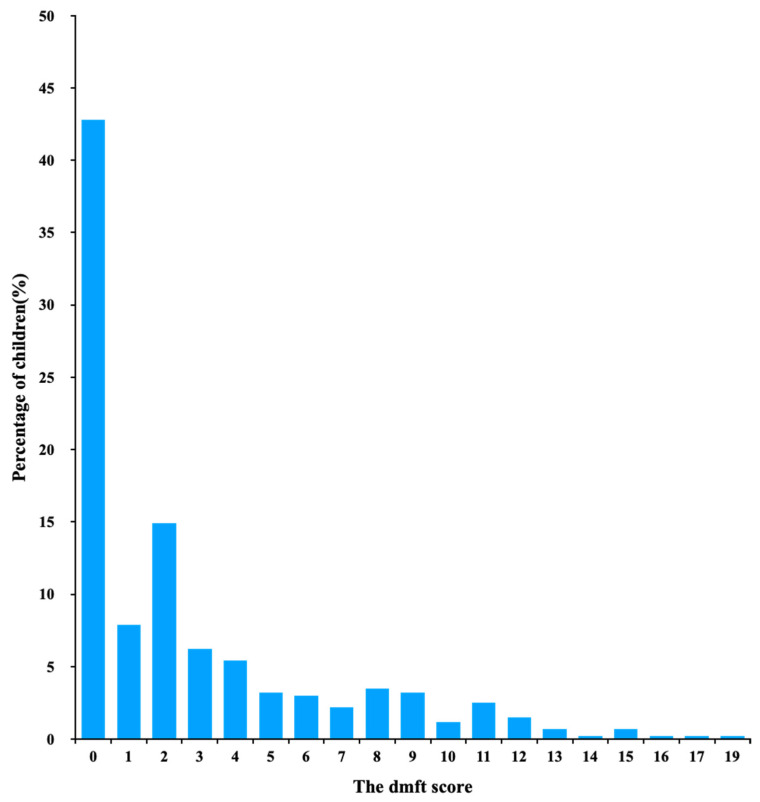
Untreated ECC index (dt) of 5-year-old children in Hong Kong (*n* = 404).

**Figure 2 ijerph-18-11934-f002:**
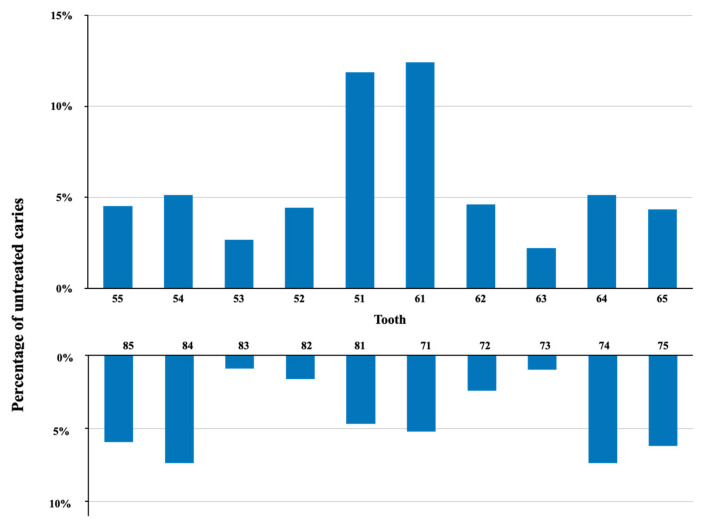
Distribution of untreated ECC index of 5-year-old children in Hong Kong (*n* = 404).

**Table 1 ijerph-18-11934-t001:** Untreated ECC prevalence and significant untreated ECC (SiUC) according to sex, birthplace and district they lived in.

Variables (*n* = 404)	Untreated ECC Prevalence	*p*-Value ^1^	Mean dt (SD)	Mean Rank of dt	*p*-Value ^2^	Mean SiUC (SD)	*p*-Value ^3^
**All children**	57%		2.8 (3.8)			7.1 (3.6)	
**Child lived in district**		**0.003**			**0.002**		**<0.001**
Hong Kong Island ^a^ (54)	43%		1.5 (2.4)	163	a < b	4.2 (2.5)	a < b, c
Kowloon ^b^ (121)	69%		3.5 (4.1)	227		8.3 (3.8)	
New Territories ^c^ (229)	55%		2.7 (3.7)	199		7.1 (3.4)	
**Born in** **mainland** **China**		0.174			0.077		**<0.001**
Yes ^a^ (45)	56%		2.6 (3.6)	198		6.8 (3.5)	
No ^b^ (357)	67%		4.1 (4.8)	229		10.1 (3.2)	a > b
**Sex**		0.480			0.256		0.369
Boys (216)	56%		2.6 (3.6)	197		6.6 (3.6)	
Girls (188)	59%		3.0 (3.9)	209		7.7 (3.5)	

^1^ Comparison of ECC prevalence; ^2^ Comparison of the mean dt; ^3^ Comparison of the mean SiUC; Bold: *p*-value *p* < 0.05.

**Table 2 ijerph-18-11934-t002:** Untreated ECC prevalence (dt > 0) and independent variables.

Variables (*n* = 404)	*n* (%)	Untreated ECC Prevalence	*p*-Value	Pairwise Comparison
**Mother’s education level**			**<0.001**	**a > b**
Secondary school or below ^a^	282 (70%)	64%		
Post-secondary school ^b^	122 (30%)	41%		
**Monthly family income (HK$)**			**<0.001**	**a > b**
Below 15,000 ^a^	99 (24%)	72%		
15,001 to 30,000 ^a^	201 (50%)	60%		
Over 30,000 ^b^	104 (26%)	39%		
**Father’s education level (*n* = 396)**			**0.004**	**a > b**
Secondary school or below ^a^	170 (67%)	62%		
Post-secondary school ^b^	130 (33%)	47%		
**Using toothpaste for tooth brushing**			**0.032**	
Yes	43 (11%)	42%		
No	361 (89%)	59%		
**Tooth brushing at least twice daily**			0.120	
Yes	263 (65%)	55%		
No	141 (35%)	62%		
**Snacking more than twice daily**			0.090	
Yes	53 (13%)	68%		
No	351 (87%)	56%		
**Parents as primary caregiver (*n* = 403)**			0.740	
Yes	287 (71%)	58%		
No	116 (29%)	56%		

*p*-value attained by chi-square test, and **bold figure**: *p*-value < 0.05.

**Table 3 ijerph-18-11934-t003:** Untreated ECC (dt) and independent variables.

Variables (*n* = 404)	Mean dt Score (SD)	Mean Rank of dt	*p*-Value	Pairwise Comparison
**Mother’s education level**			**<0.001**	**a < b**
Secondary school or below ^a^	3.3 (3.9)	219		
Post-secondary school ^b^	1.7 (3.1)	164		
**Monthly family income (HK$)**			**<0.001**	**a > b > c**
Below 15,000 ^a^	4.1 (4.4)	240		
15,001 to 30,000 ^b^	2.8 (3.7)	207		
Over 30,000 ^c^	1.4 (2.6)	157		
**Father’s education level (*n* = 396)**			**0.004**	**a > b**
Secondary school or below ^a^	3.1 (3.9)	210		
Post-secondary school	2.1 (3.4)	176		
**Snacking more than twice daily**			**0.040**	
Yes	3.9 (4.4)	232		
No	2.6 (3.6)	198		
**Using toothpaste for tooth brushing**			**0.029**	
Yes	1.7 (2.8)	167		
No	2.9 (3.8)	207		
**Tooth brushing at least twice daily**			0.072	
Yes	2.7 (3.8)	195		
No	3.0 (3.6)	216		
**Parents as primary caregiver (*n* = 403)**			0.920	
Yes	2.8 (3.8)	202		
No	2.7 (3.6)	201		

*p*-value attained by Mann–Whitney U test or Kruskal–Wallis H test, and **bold figure** means *p*-value less than 0.05.

**Table 4 ijerph-18-11934-t004:** Final model of zero-inflated negative binomial analysis of untreated ECC and risk factors.

Negative Binomial Portion of the Zero-Inflated Negative Binomial Analysis				
Variables (*n* = 401)	Incidence Risk Ratio	*p*-Value ^#^	95% Confidence Interval	Pairwise Comparison
**Monthly family income (HK$)**		**0.004**		**a > c**
* Below 15,000 ^a^				
15,001 to 30,000 ^b^	0.801	0.093	(0.619, 1.037)	
Over 30,000 ^c^	0.589	**0.005**	(0.408, 0.852)	
**Zero-Inflated Portion of the Zero-Inflated Negative Binomial Analysis**				
**Variables** **(*n* = 401)**	**Odds Ratio**	** *p* ** **-Value ^#^**	**95%** **Confidence Interval**	**Pairwise Comparison**
**Monthly family income (HK$)**		**0.011**		**c > a**
* Below 15,000 ^a^				
15,001 to 30,000 ^b^	1.770	0.123	(0.857, 3.655)	
Over 30,000 ^c^	3.455	**0.003**	(1.524, 7.830)	
**Mother’s education level**		**0.007**		
* Secondary school				
Post-secondary school	2.211		(1.238, 3.949)	
**Using toothpaste when brushing**		**0.044**		
* No				
Yes	2.362		(1.022, 5.463)	

^#^ Bold figure: *p*-value < 0.05; * Reference group.

## Data Availability

The datasets generated and/or analyzed during the current study are available from the corresponding author upon reasonable request.

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
