# Peer review of "Prevalence of Untreated Early Childhood Caries of 5-Year-Old Children in Hong Kong: A Cross-Sectional Study"

_ijerph, 2021, doi:10.3390/ijerph182211934_

Round 1
Reviewer 1 Report
Thank you for the opportunity to review the work. its very interesting, has been well done and provides insightful information. However, below are some comments that might be helpful.
1) Introduction - It is great to see the extent of surveillance conducted among young children under 5 yrs of age in HongKong. However, there is mention of only one program in 2019 to address ECC. Were there no preventive/theraputic programs conducted before that? Mention of some would be useful for context. Table 1 is also not required in my opinion (or can be a supplementary table). The content of the table can be incorporated into the introduction to better inform the objectives of the study.
2) Methods- It would help to frame the methods as follows : i) Study participants, ii) sampling methods, iii) sample size calculation, iv) Questionnaire survey, v) clinical examination, vi) statistical analysis, vii) ethical considerations.
Please add eligibility criteria (has been mentioned in the discussion section - lines 214-217).
In clinical examination, its unclear who did the second clinical exam to test for intra-examiner agreement. Was it the dental assistant who conducted the clinical exam on the 10% random sample or the dentist. (move lines 239 - 243)
In the sample size calculation, did you recruit children from daycares/schools? If yes, did you randomly select schools from the three districts? Additional clarifications would help. (section relevant to this is in the discussion section, lines 219-230, 232-237)
In Statistical analysis, please explain the SiUC and what it means for the analysis. Please also add the independent variables of interest that were initially accounted for in the regression modeling.
3) Results - nicely done.
Please add a table that presents the socio-demographic and other variables' distribution in the sample population (for both children and parents).
One question- - do you think that the fact tat 89% of the children's surveyed used a toothpaste makes the variable significant when comparing with those who did not use toothpaste? the small sample size may have impacted the association.
4) Discussion -
Use of STROBE guidelines should be moved to the methods section.
Lines 259 - 270 - the findings align with general consensus that eECC is more common in upper front and lower posterior teeth. Please make that explicit. The way it has been presented seems like this is not an ordinary finding.
you have mentioned the club charities trust that supports dental outreach - has there been any evaluation of this outreach program to gauge its effectiveness. Please add relevant information (if there are no evaluations, please mention). Some additional recommendations should be provided to align your findings with the high prevalence of ECC. Example of Singapore is helpful - however, please provide some detail on how Singapore's govt helped shift the needle. Not sure if the UK example is relevant here.
Thank you
Reviewer 2 Report
This is an interesting paper that aimed to investigate untreated early childhood caries (ECC) and its 10 associated factors among 5-year-old children in Hong Kong. I have few minor comments for authors.
- What was the duration of research period, clinical examination to be specific? Was the clinical examiner recalibrated during the study period?
- This study was based on multistage sampling method. How did author take this into account during statistical analyses?
- In the result section, the authors have compared the Mean/proportion between groups and reported the result as “significant”/ “no significant” based on the p-values generated from tests. I suggest removing these terms throughout the text in the result section. Furthermore, I recommend reading the following paper:
Rafi & Greenland (2020): Semantic and cognitive tools to aid statistical science: replace confidence and significance by compatibility and surprise. https://doi.org/10.1186/s12874-020-01105-9.
Reviewer 3 Report
The cross-sectional study evaluates the untreated prevalence of early childhood caries (ECC) among 5-year-old children in Hong Kong and the causal relationship between them and a series of socio-demographic factors, snacking behaviours, respectively oral hygiene practice. Overall, the study is clearly and detailed presented, the topic is interesting, but I have some major concerns. Please see the comments below.
- In the introductory part I recommend emphasizing the determining etiological factor of EEC, which is represented by the frequent sugar consumption in an environment of enamel adherent bacteria and also the detailing of the mechanism by which it leads to sustained acid production and consequent demineralization of the tooth structure.
- Also in the introduction, I would point to a more scientific and comprehensive approach when exposing the complications of oral health with respect to the untreated ECC. These are not just about toothache and infection (lines 36-37), being about more immediate and long-term consequences such as the higher risk of new carious lesions, malocclusion with various clinical implications (e.g. deep overbite, midline deviation, excessive overjet, anterior cross-bite, mal-alignment), extraction of primary teeth, and not only.
Please remove the fragment Untreated ECC has consequences and causes pain and infection from line 65, as it is identical to the one on lines 36-37.
- Were the children/the children’s parents incentivized to participate in the study?
- The description of research tools used [chapter 2.2 Questionnaire survey] requires completing. In my opinion, the questionnaire for parents of children should be found at least as a supplementary file to this article, even if it has been used in previous oral health surveys conducted by you.
Furthermore, regarding the 2016 questionnaire (Chen, KJ; Gao, SS; Duangthip, D .; Li, SKY; Lo, ECM; Chu, CH Dental caries status and its associated factors among 5- year-old Hong Kong children: a cross-sectional study. BMC Oral Health 2017, 17, doi: 10.1186 / s12903-017-0413-2), you mentioned there other twenty-one multiple-choice questions adapted from previous surveys for parents of Hong Kong preschool children about the child's parent's dental knowledge, to which you gave scores, depending on the answer option chosen. These items are no longer mentioned, however, when you briefly describe the questionnaire currently applied. Please clarify.
- Lines 129-130- You mention here a number of 231 children who have untreated ECC, while in the abstract the number is 230, see row 20.
- On lines 170-171 you wrote: We also found that most of the children (361/404, 89%) brushed their teeth with toothpaste. This statement contradicts the data in Table 3, where toothpaste for tooth brushing is NOT used by 361 of the children.
- Nowhere in the description of the questionnaire, or elsewhere in the Material and Method section, is being mentioned how the data on dental attendance or experience of the investigated children were collected, although they later appear in the Results, between lines 173 and 175, and surprisingly to the conclusions, on line 315. I suggest that these parameters, the way they were evaluated to be found in the Material and Method and in the Results, including in table 3. Furthermore, there is no clear difference between no experience of dental attendance (line 174) and never visited a dentist (line 175).
- On lines 181-182 you wrote: The Mann – Whitney U and Kruskal – Wallis H tests showed a higher dt distribution for children who…. ate sugary snacks more than twice daily (p = 0.040)…, a statement that contradicts the data in Table 4 on this topic.
- A previous article by you (Chu, CH; Ho, PL; Lo, ECM Oral health status and behaviors of preschool children in Hong Kong. BMC Public Health 2012, 356 12, 1-8, doi: 10.1186 / 1471-2458- 12-767.) on the same topic, states in the Discussion section: The dental caries experience of the preschool children in this survey is higher than that of the preschool children in a recent survey conducted in Singapore… .. It should be noted that Hong Kong and Singapore are similar in terms of economic development and that both cities have implemented water fluoridation…
Please carefully analyze the fragment from your current article, between lines 299 and 302: The prevalence of ECC in Singapore is lower than that in Hong Kong… .Singapore and Hong Kong have similar economic development. They fluoridate their water and have a similar dental care system. Basically you provide the same information to the discussion part, while each new clinical study, even if methodologically and of the proposed purpose is similar to the previous ones, should bring its part of novelty, relevance and a different approach to scientific level. Please take care of this issue carefully.
- In the Discussion section, between lines 214 and 218, you wrote: We selected the children by random sampling in baseline, and they did not receive the dental care outreach service. To evaluate the dental outreach service, we compared their ECC status to those children of the same age who had received the dental care outreach service after 36 months.
The whole paragraph is confusing. What do you mean by the study compared to those who received dental care for 36 months? Is it still a study of yours? I recommend avoiding confusions, clearer expressions, indicating the logical thread of those exposed and indicating the bibliographic reference. Make it clear for the readers to understand it in a few sentences.
- The discussion part, as well as the conclusion part, although structurally correct, is very similar to your previous articles, on the same topic. It would be desirable that at least one element of novelty be brought to light that attracts the attention of readers.
- The fact that the untreated prevalence of ECC among 5-year-old children in Hong Kong in 2021 not only remains high but is higher than those surveys reported in the past two decades (Line 294-295) is important to explain and also to point out as an increased treatment need among these population category in Hong Kong, given the importance of keeping as intact as possible on the dental arches of some of the temporary teeth even up to the age of 13 years.
Reviewer 4 Report
Introduction
Page 2 line 65: “Untreated ECC has consequences and causes pain and infection.” This statement repeats the information that is included in lines 36-38. I suggest to remove it from the main text.
Material and methods
2.3 Clinical examination
It is pointed out that a dentist performed the clinical examination for the diagnosis of dental caries, with no specification regarding whether a calibration was developed to accomplish this procedure. Then, it is indicated that a dental assistant performed the examination in duplicated without notifying the dentist. Was a calibration conducted for the dental assistant? What was the kappa value for the inter-examiners evaluation?
Results
Page 3 line 132. “Kappa value for intra-examiner agreement” should be placed in the material and methods section of the study, specifically within clinical examination.
Page 4 line 139. It is highlighted the prevalence of ECC in the lower central incisor, would it be possible to provide an explanation of why this result is relevant?
Page 5 lines 150-153. The authors compare the prevalence of untreated ECC in children that were and were not born in mainland China. However, in the material and methods section was specified that the survey was only conducted in three different districts of Hong Kong. Does this comparison correspond to what is reported in this research? Or does it refer to a different study? Was any district included in the comparison between children born and not born in mainland China?
Discussion
Page 8 line 213-214. It says “We followed the Statement of Strengthening the Reporting of Observational Studies in Epidemiology (STROBE) in our reporting of this survey.” This statement should be placed in the material and methods section due to it refers to the guidelines that are followed when performing observational studies.
Page 8 lines 216-218. The authors indicate that they “…compared their ECC status to those children of the same age who had received the dental care outreach service after 36 months.” I consider that this statement should be included as inclusion criteria in the material and methods section. Likewise, there should be included the exclusion and elimination criteria.
Page 9 lines 278-280. It says “Hong Kong is part of China, and over the past 20 years, an 278 average of 131 new immigrants travel to Hong Kong every day, mainly for family reunions.” I consider that this information is irrelevant for the study, thus, I suggest to delete it.
The discussion seems to be focused on explaining each employed method in the development of the study. However, this should be focused exclusively on the comparison of the results presented in the previous surveys and the survey conducted in the current study, as is performed from line 259. I suggest to reorganize this section to improve the included content.
Conclusion
The conclusion does not widely reflect the findings of the current study. It is necessary to point out how the present study contributes to the ECC diagnosis in the specific studied areas, the limitations and strenghts that were observed, as well as the different aspects of ECC that might be studied in further research. I suggest to complement the content of this section.
Round 2
Reviewer 3 Report
Dear Authors,
Revisions have been made mainly accordingly, thanks for that. On the discussion side, another possible explanation for the alarming increase of untreated ECC in 2021 may be the interruption of kindergarten attendance due to the COVID 19-pandemic, with the change in the frequency of snaking by children who have stayed home for a long time. At this point you may take into account this reasoning too, knowing exactly how things were related to the Hong Kong lockdown.
